# A Fail-Operational Control Architecture Approach and Dead-Reckoning Strategy in Case of Positioning Failures

**DOI:** 10.3390/s20020442

**Published:** 2020-01-13

**Authors:** Jose Angel Matute-Peaspan, Joshue Perez, Asier Zubizarreta

**Affiliations:** 1Tecnalia Research & Innovation, 48160 Derio, Spain; joshue.perez@tecnalia.com; 2Department of Automatic Control and Systems Engineering, University of the Basque Country (UPV/EHU), 48013 Bilbao, Spain; asier.zubizarreta@ehu.eus

**Keywords:** fail-operational systems, fall-back strategy, automated driving

## Abstract

Presently, in the event of a failure in Automated Driving Systems, control architectures rely on hardware redundancies over software solutions to assure reliability or wait for human interaction in takeover requests to achieve a minimal risk condition. As user confidence and final acceptance of this novel technology are strongly related to enabling safe states, automated fall-back strategies must be assured as a response to failures while the system is performing a dynamic driving task. In this work, a fail-operational control architecture approach and dead-reckoning strategy in case of positioning failures are developed and presented. A fail-operational system is capable of detecting failures in the last available positioning source, warning the decision stage to set up a fall-back strategy and planning a new trajectory in real time. The surrounding objects and road borders are considered during the vehicle motion control after failure, to avoid collisions and lane-keeping purposes. A case study based on a realistic urban scenario is simulated for testing and system verification. It shows that the proposed approach always bears in mind both the passenger’s safety and comfort during the fall-back maneuvering execution.

## 1. Introduction

In the last decade, Automated Driving Systems (ADS) has shown significant advances, mainly from the acquisition, perception, control and actuation point of view [1]. Several important developments have been achieved and mentioned in the latest European Commission reports [2], where the challenges on communication technologies and cyber-security, on-board sensors capacities, infrastructure requirements, mobility concepts, and city contexts are playing an active role for sustainable urban transportation developments.

ADS obtain information about the surroundings from different sensors, such as cameras, differential global positioning systems (GPS), Light Detection and Ranging (LiDAR), and Radio Detection and Ranging (RaDAR) systems [3]. Perception tasks are critical for increasing the level of automation of ADS developments, as environment recognition in any scenario, including lighting and weather conditions, should be assured. Moreover, their fail-operational operation during autonomous mode is crucial to ensure passenger safety, as sensor and perception errors can be easily propagated to decision and control stages in different maneuvers, causing fatal accidents [4].

Some authors have considered sensor data fusion for more robust performance on different contexts: obstacles detection [5], perception of the environment [6], localization [7], and Traffic Sign Detection and Recognition [8]. A detailed description of the most popular methods and techniques for performing data fusion is presented in [9], where the author concludes that the appropriate technique to be implemented depends on the type of problem. In the automotive field, the Bayesian approach, extended and unscented Kalman filters (UKF) are mostly used [10,11,12]. However, these techniques depend mainly on the information directly from on-board sensors without any fall-back strategy.

Currently, the most widely used global localization approaches involve Global Navigation Satellite Systems (GNSS) such as GPS and Galileo [13]. The implemented devices even can implement differential GPS approaches, which have become affordable in recent years, or Inertial Measurement Units (IMU) [7], which may be fused with GNSS data to provide more reliable data. Although this approach works properly in open scenarios such as highways, in urban environments, their localization accuracy is not guaranteed for ADS [13]. Hence, a fail-operational positioning system, which also uses the dynamic model of the vehicle, is required to increase the accuracy. Moreover, the implemented ADS need to have a Dynamic Driving Task (DDT) fall-back strategy approach to be executed when the positioning system fails [14]. Among the different proposed strategies Table 1 summarizes the most important DDT fall-back strategies proposed.

However, a better assessment of fail-operational strategies for the functions of the dynamic driving task (DDT) is needed to achieve higher levels of automation on ADS (SAE J3016 [23]). This work is focused on this area, and its main contribution of this work is a fail-operational strategy approach considering positioning failures in a last available device, implemented within a general control architecture for automated vehicles. In brief, the improvements presented in this work are:A fail-operational positioning system that comprises a UKF, a virtual sensor, and a monitor system, capable of remaining operative from degraded to total failure of the position reception and warns for fall-back triggering.A real-time trajectory planner that defines the lateral and longitudinal references for the DDT fall-back in degraded mode to achieve a minimal risk condition avoiding rear-end collisions.A vehicle motion control that executes the planned trajectory, including a lateral reference constraint avoiding undesirable lane departures.A case study resembles a real urban scenario demanding a DDT fall-back strategy due to a major positioning failure, working with minimum sensor interface bringing the vehicle to a safe place.

The rest of the paper is organized as follows. In Section 2, the fail-operational control architecture is detailed, explaining each module. An overview of the fail-operational positioning system is presented in Section 3, where the vehicle model, cornering stiffness estimation and the adaptive UKF are the main contributions. Section 4 describes the DDT fall-back strategy proposed for the decision stage, considering a real-time trajectory generation, rear-end collision avoidance and vehicle motion control. In Section 5, a description of the scenario, the test platform and the parameter of configuration for the UKF and MPC are presented. Results and discussion are explained in Section 6. Finally, some remarks and conclusions are presented in Section 7.

## 2. Fail-Operational Control Architecture

The proposed work has been developed in the framework of the AutoDrive Project [24]. This program targets the development of SAE Level 4 [23] automated driving capabilities. More precisely, a highly automated driving bus to carry passengers in an urban scenario with mixed traffic conditions.

Please note that the required automation level must include, moreover the lateral and longitudinal motion control, a complete Object and Event Detection and Response (OEDR) system, and the capability to be robust enough to support fail-operational operation [25], which is the focus of this work.

The control architecture proposed to achieve this goal is depicted in Figure 1, covering seven stages of those suggested by [1] required for ADS developments (Database, Acquisition, Perception, Supervisor, Decision, Control, and Actuation). This architecture allows the verification of the DDT fall-back strategy after the occurrence of a positioning system failure.

In the next subsections, the different stages that compose the proposed architecture are detailed.

### 2.1. Database

The database is composed of a list of waypoints that contain relevant information of the fixed route, such as global axis coordinates (X,Y), orientation angles (ψ) and velocity limits ($v_{x}$). Additionally, information related to safe-parking places is included, considering three different cases: stop not permitted, stop on-lane permitted and stop on-shoulder available. These waypoints will be illustrated in the case study in Section 5.

### 2.2. Acquisition

The acquisition system interface provides two features: the vehicle surrounding recognition, and the vehicle global position on the route.

For the first feature, a sensor interface is used, which provides relevant information about the surroundings. This will be processed by the perception stage simulating information from road borders and objects. Please note that in the present work’s framework the sensor interface will be idealized to provide information from 360 degrees around the vehicle, with a maximum detection radius. Elements outer the range will not be detected.

To define the position of the vehicle in the scenario, a global navigation satellite system integrated with an inertial navigation system (GNSS-INS), odometer and steering sensors are considered. These devices provide the vehicle position (X,Y), front-wheel angle (δ) and inertial parameters as orientation (ψ), acceleration ($a_{x}$) and velocity ($v_{x}$).

Please note that commercial GNSS-INS interfaces present noise and signal quality reductions. Hence, the failure of this sensor must be handled by the proposed fail-operational positioning system and fall-back strategy.

### 2.3. Perception

The information provided by the acquisition blocks is used to detect the road borders and objects within the sensor range. These are estimated in coordinates relative to the vehicle. This way, a 4-m road width is considered, so that the road borders are placed at 2 m of lateral distance from the center-lane (X,Y). The lateral distances are estimated from the vehicle’s position to the left ($e_{L}$) and right ($e_{R}$) borders. On the other hand, an object list is provided considering relative distances and velocities.

### 2.4. Supervisor. Fail-Operational Positioning System

The supervisory system is continuously monitoring the positioning accuracy and the status of the positioning sensor devices (GNSS-INS) of the acquisition blocks.

When the vehicle performance is highly compromised or a relevant sensor failure is detected, a fail-operational strategy is activated. Using the data from the positioning sensors (GNSS-INS, odometer and steering sensors), a virtual positioning sensor is switched on and employed to perform the fall-back strategy that leads the test platform to a safe state. This one of the contributions of this work, and its development is broadly detailed in Section 3.

### 2.5. Decision

The decision module creates the trajectories to be followed by the automated vehicle and is integrated by the trajectory planner and a collision-avoidance system.

The trajectory planner, in the case of normal operation, will generate optimum trajectories for a specified scenario. However, in the case of system failure, such as the case study analyzed in Section 5, the trajectory planner will adjust the original route. Using the information of the safe-parking places from the database, the planner will modify the route to achieve the nearest safe state. This is an important contribution of this work and will be covered in Section 4.1.

On the other hand, the collision-avoidance system is focused on avoiding rear-end collisions with previous vehicles or objects. Please note that the objects detected by the perception system are first analyzed to evaluate if they are within the trajectory to be followed. Therefore, objects located outside the road borders do not represent a collision risk and adjustments are not necessary over the original trajectory, while objects within the trajectory will require adjustment of the trajectory by maintaining a safe distance to the objects ahead. The proposed approach, which will be detailed in Section 4.2, can work not only in normal operation, but also when there exists a degraded operation due to failure in the positioning sensors.

### 2.6. Control

The trajectory references estimated in the decision stage are followed by the vehicle motion control, providing reliable inputs to the vehicle interface. The velocity ($v_{x}$), acceleration ($a_{x}$) and jerk ($j_{x}$) are usually considered to be the main state parameters to control the longitudinal vehicle motion behavior. The position in global coordinates (X,Y) and the yaw angle of the vehicle (ψ) are commonly used for lateral and angular vehicle motion control, respectively. Additional relevant state parameters can be included for control improvement as the lateral error concerning the route’s center-lane ($e_{y}$). A detailed explanation of this stage is presented in Section 4.3.

### 2.7. Actuation

The actuation module receives information from the control stage. Its task is to move the actuator of the automated vehicle, which means, steering wheel and pedals. Characterization of the actuators is the proposed architecture, as presented in [26].

## 3. Fail-Operational Positioning System

The proposed approach uses a sensor fusion strategy that combines the information of the GNSS-INS, odometer and steering sensor to provide positioning data even in degraded circumstances. Moreover, it integrates a quality monitor that allows detecting the failure of the sensor. The overall architecture of the proposed fail-operational positioning system is depicted in Figure 2.

This strategy first uses the current values of velocity ($v_{x}$) and front-wheel angle (δ) to interpolate the front ($C_{\alpha_{f}}$) and rear ($C_{\alpha_{r}}$) cornering stiffness from database values. This data will be used in a second step, where an adaptive UKF is employed to attenuate the accuracy lacking on the GNSS-INS positioning measurement. This is a problem regularly faced in urban environments due to obstructions in the line-of-sight to the satellites [12].

When a relevant performance failure is detected, a virtual positioning sensor is activated. It makes use of the last position acquired by the UKF system, the estimation of the cornering stiffness, and the data provided by the odometer sensors and steering wheel sensor to estimate positioning to perform dead reckoning and perform a safe state on the route.

The detection of a degraded condition and critical failure is performed by a positioning monitor, which selects the source of the positioning data to provide a fail-operational response, and informs the decision stage about the failure.

### 3.1. Vehicle Model and Cornering Stiffness Estimation

The proposed fail-operational positioning system requires a vehicle model to implement both the lateral vehicle motion control and UKF. In this section, the model used for the development of the UKF is briefly detailed.

#### 3.1.1. The Kinematic and Dynamic Vehicle Models

The lateral motion of a vehicle can be estimated as well as controlled employing simplified vehicle models, being this a technique that reduces the computational effort for real-time implementations while providing enough accuracy for control purposes [27]. For velocities at less than 3 m/s the lateral forces on the tires can be neglected, and the vehicle motion can be calculated entirely on geometric relationships of *X*, *Y* and ψ [28,29]. Above 3 m/s, the assumption of no lateral forces on the tires begins to be compromised, as the lateral vehicle motion is affected by its dynamics being necessary to take into consideration a more complex model to improve results [30]. In these cases, a mix of simplified bicycle models for lateral vehicle dynamics (Figure 3) provides a good accuracy vs complexity relationship.

In Figure 3, *V* represents the velocity at the center of gravity (CG) and β is the vehicle slip-angle concerning the longitudinal axis of the vehicle (*x*). The lateral tire forces on the front and rear wheels ($F_{f}$, $F_{r}$) are strongly affected by the cornering stiffness of each tire ($C_{\alpha_{f}}$, $C_{\alpha_{r}}$), the slip-angle of each tire ($\theta_{f}$, $\theta_{r}$) and the δ of the front wheel. The lateral translation motion and the yaw dynamics for the model can be written as,
(1a)$$\begin{array}{c} m\left(\dot{v_{y}}+\dot{\psi }v_{x}\right)=2C_{\alpha_{f}}\left(\delta -\theta_{f}\right)-2C_{\alpha_{r}}\theta_{r} \end{array}$$
(1b)$$\begin{array}{c} I_{z}\ddot{\psi }=2l_{f}C_{\alpha_{f}}\left(\delta -\theta_{f}\right)+2l_{r}C_{\alpha_{r}}\theta_{r} \end{array}$$
where *m* is the mass, $\dot{v_{y}}$ is the lateral acceleration, $\dot{\psi }$ is the yaw rate, $v_{x}$ is the longitudinal velocity, $I_{z}$ is the moment balance around the *z* axis of the vehicle, $\ddot{\psi }$ is the yaw acceleration and, $l_{f}$ and $l_{r}$ are the front and rear wheel distance from the CG. The slip angles of each tire ($\theta_{f}$ and $\theta_{r}$) can be calculated as,
(2a)$$\begin{array}{c} \theta_{f}=\mathrm{atan}\frac{v_{y}+l_{f}\dot{\psi }}{v_{x}} \end{array}$$
(2b)$$\begin{array}{c} \theta_{r}=\mathrm{atan}\frac{v_{y}-l_{r}\dot{\psi }}{v_{x}} \end{array}$$

In this work, a dynamic bicycle model approach is implemented for filtering and positioning. On the other hand, a kinematic bicycle model approach is employed for vehicle motion control.

#### 3.1.2. Cornering Stiffness Estimation

Using the dynamic bicycle model defined in Equation (1) it is possible to estimate the cornering stiffness coefficients $C_{\alpha_{f}}$ and $C_{\alpha_{r}}$. Please note that most parameters are easy to obtain in real applications [31]. This way, if a state-space representation is used,
(3)$$\left[\begin{array}{c} C_{\alpha_{f}}\\ C_{\alpha_{r}} \end{array}\right]={\left[\begin{array}{cc} 2(\delta -\theta_{f}) & -2\theta_{r}\\ 2l_{f}\left(\delta -\theta_{f}\right) & 2l_{r}\theta_{r} \end{array}\right]}^{-1}\left[\begin{array}{c} m(\dot{v_{y}}+\dot{\psi }v_{x})\\ I_{z}\ddot{\psi } \end{array}\right]$$

An open-loop test method for determining the steady-state circular driving behavior described in [32] (ISO 4138) is employed for the cornering stiffness estimation at constant steering wheel angles and velocities. Using this procedure, a cornering stiffness map can be generated and integrated into the control architecture, to implement a cornering stiffness estimator as input to the UKF (Figure 2), so that for any $v_{x}$ and δ the coefficients $C_{\alpha_{f}}$ and $C_{\alpha_{r}}$ can be derived. Results of these tests for the case study analyzed in this work are presented in Section 5.3.

Please note that the use of an off-line cornering stiffness estimator implies that the open-loop tests must cover the whole range of $v_{x}$ and δ to be performed by the test vehicle within the ODD. Although this can be implemented for on-line estimations, this procedure helps to fix values when necessary avoiding some singularities in circumstances, as no lateral accelerations [33].

### 3.2. Adaptive Unscented Kalman Filter

An adaptive Unscented Kalman Filter (UKF) is used to attenuate the errors introduced in the GNSS-INS positioning measurement when the satellite signal quality is reduced. In contrast to other Kalman filtering techniques, the UKF frequently provides a lower estimation error and is preferable for implementations in automated driving applications [10]. In this sense, a UKF-based approach capable of adapting the measurement noise covariance matrix is presented here, this is an adaptive UKF.

The development of the UKF requires the space-state transition model of the vehicle detailed in Section 3.1, which can be defined as,
(4)$$\left[\begin{array}{c} \dot{X}\\ \dot{Y}\\ \dot{y}\\ \dot{v_{y}}\\ \dot{\psi }\\ \ddot{\psi } \end{array}\right]=\left[\begin{array}{cccccc} 0 & 0 & 0 & -\sin\psi & \frac{v_{x}\cos\psi }{\psi } & 0\\ 0 & 0 & 0 & \cos\psi & \frac{v_{x}\sin\psi }{\psi } & 0\\ 0 & 0 & 0 & 1 & 0 & 0\\ 0 & 0 & 0 & -\frac{2C_{\alpha_{f}}+2C_{\alpha_{r}}}{mv_{x}} & 0 & -v_{x}-\frac{2C_{\alpha_{f}}l_{f}-2C_{\alpha_{r}}l_{r}}{mv_{x}}\\ 0 & 0 & 0 & 0 & 0 & 1\\ 0 & 0 & 0 & -\frac{2l_{f}C_{\alpha_{f}}-2l_{r}C_{\alpha_{r}}}{I_{z}v_{x}} & 0 & -\frac{2{l_{f}}^{2}C_{\alpha_{f}}+2{l_{r}}^{2}C_{\alpha_{r}}}{I_{z}v_{x}} \end{array}\right]\left[\begin{array}{c} X\\ Y\\ y\\ v_{y}\\ \psi \\ \dot{\psi } \end{array}\right]+\left[\begin{array}{c} 0\\ 0\\ 0\\ \frac{2C_{\alpha_{f}}}{m}\\ 0\\ \frac{2l_{f}C_{\alpha_{f}}}{I_{z}} \end{array}\right]\delta$$
where *X*, *Y*, $v_{y}$, ψ and $\dot{\psi }$ are parameters obtained from the GNSS-INS interface. The $v_{x}$ and δ are parameters received from the odometer and steering angle sensor interface. A linear relationship between the steering angle and the front-wheel angle is employed to obtain the current value of δ. The stiffness coefficients $C_{\alpha_{f}}$ and $C_{\alpha_{r}}$ are obtained through the procedure described in Section 3.1.2.

The process noise covariance matrix ($Q_{n}$) in a vehicle model is suggested to be calculated as the propagation of each value per time step [11], in this sense, gathering the standard deviation of parameters from the test vehicle circulating in normal conditions helps to determine $Q_{n}$.

The measurement noise covariance matrix ($R_{n}$) is mainly associated with the accuracy of the acquisition devices. These can be extracted from commercial GNSS devices data-sheet.

### 3.3. Virtual Positioning Sensor

When a GNSS failure event occurs (lower signal quality or total disconnection), a virtual positioning sensor is used to provide an indirect position measurement by combining information from the remaining physical sensors.

The velocity from odometer ($v_{x}^{odo}$), the lateral velocity from the filter ($v_{y}^{ukf}$), and the yaw angle obtained due a discrete integration from the filter yaw rate measure ($\psi^{int}$), are considered to estimate the vehicle velocities in global coordinates ($\dot{X}$,$\dot{Y}$). A state-space representation is described as,
(5)$$\left[\begin{array}{c} \dot{X}\\ \dot{Y} \end{array}\right]=\left[\begin{array}{cc} \cos\psi^{int} & -\sin\psi^{int}\\ \sin\psi^{int} & \cos\psi^{int} \end{array}\right]\left[\begin{array}{c} {v_{x}}^{odo}\\ {v_{y}}^{ukf} \end{array}\right]$$

The obtained velocities are consequently integrated to obtain *X* and *Y*. The last available values before the failure for *X*, *Y* and ψ are considered to be the initial values for the newly integrated parameters. The remaining available parameters as $v_{y}$ and $\dot{\psi }$ are combined with the indirect estimations to maintain the same structure information sent by the UKF.

### 3.4. Positioning Monitor

The monitor role is to continuously evaluate the positioning quality of the GNSS. In case of a very poor positioning accuracy (quality below 2 in Table 2) or a catastrophic failure (e.g., power supply unavailable), the monitor will instantly switch the information received from the UKF to the one received from the virtual sensor. The output state parameters are combined with a failure tag (1/0) to inform this status to the decision stage so that a degraded condition and proper action taken.

## 4. Fall-Back Strategies Implementation in the Decision Stage

The fail-operational positioning system provides information on the vehicle position and the existence of a failure to the decision stage of the control architecture depicted in Figure 1. In this section, the fall-back strategy, which includes both the trajectory planner and the collision-avoidance subsystems, will be detailed.

### 4.1. Real-Time Trajectory Planner

In normal operation, a fixed route is planned off-line based on Bezier and feasible curvatures generation procedure [34]. The velocities are limited considering the curvatures along the route [26] defining bounds for lateral and longitudinal accelerations bearing in mind the passenger comfort [35].

In case of failure, a DDT fall-back strategy starts, and the trajectory planned is modified to achieve a degraded driving mode. The velocity is instantly reduced to a degraded value, to avoid lateral displacements in vehicle motion control while dead-reckoning is performed and maintained until the vehicle is located over a safe-parking spot, where the vehicle stops. The path is not modified until a safe-parking space is available.

The strategy for a degraded velocity ($v_{x}^{degr}$) is depicted in Figure 4a. After the failure, a start distance ($d_{start}$) is defined to reduce the speed at degraded deceleration ($a_{x}^{degr}$), as a sudden reduction could affect negatively on the motion controller, producing undesirable and uncomfortable responses. The same procedure is repeated to stop once the vehicle is located over the emergency shoulder.

The strategy for a degraded path is presented in Figure 4b. After the failure, the planned path is maintained until a safe-parking place becomes available (${\left[X,Y\right]}^{start}$), at this point, the planned route is moved perpendicularly based on a predefined lateral velocity ($v_{ey}$) to a proper distance in the emergency shoulder ($d_{ey}$).

The degraded path reference is estimated to displace laterally from the original route faster than the vehicle’s capabilities, therefore the absolute value of the lateral error increases and decreases during the lane-change maneuver. The $d_{ey}$ magnitude helps to predict when the vehicle goes out the main route ($d_{ey}>0.64$ m) and afterward is located enough on the emergency shoulder ($d_{ey}<0.16$ m), finally permitting reduction of the degraded velocity to zero. A flowchart of the real-time trajectory planning is depicted in Figure 5. The practicability of this methodology is discussed in Section 6.1.

### 4.2. Rear-End Collision Avoidance

In both normal and degraded operation, the automated vehicle implements a rear-end collision-avoidance system using the data provided by the fail-operational positioning system and a detection system of the objects ahead.

For that purpose, a Model Predictive Control (MPC) approach has been implemented, generating the velocity references to be followed by a low-level control, attempting to maintain a safe relative distance ($d_{r}$) and velocity ($v_{r}$) from objects ahead on-route. A one-dimensional kinematic model is considered to model the longitudinal vehicle motion,
(6)$$\left[\begin{array}{c} \dot{v_{x}}\\ \dot{a_{x}}\\ \dot{d_{r}}\\ \dot{v_{r}} \end{array}\right]=\left[\begin{array}{c} a_{x}\\ j_{x}\\ v_{t}-v_{x}\\ a_{t}-a_{x} \end{array}\right]$$
where $j_{x}$ is the longitudinal jerk, $v_{t}$ is the target object velocity and $a_{t}$ is the target object acceleration.

The state parameters η = [$v_{x}$
$d_{r}$
$v_{r}$] are optimized in the entire prediction horizon (*H*). The references for $v_{x}$ are defined by the planned velocity discussed in Section 4.1. The reference values for $d_{r}$ and $v_{r}$ are defined as,
(7a)$$\begin{array}{c} d_{r}^{ref}=d_{r}^{min}+v_{x}t_{hw} \end{array}$$
(7b)$$\begin{array}{c} v_{r}^{ref}=0 \end{array}$$
where $d_{r}^{min}$ is the minimum safety distance at 5 m, and $t_{hw}$ is a headway time equals to 1 s.

The state parameters weighting matrix $Q_{w}$ changes according to a $d_{r}$-$v_{r}$ diagram [28], which determines the operation mode to perform velocity or headway control in case an object detection.

The maximum deceleration permitted ($a_{x}^{ref}$) changes also with the operation mode, being this an important value to properly perform a longitudinal vehicle motion control. Lower and upper bounds are considered to maintain properly a safe distance from objects ahead as 5 m $<d_{r}<$ 50 m.

As current detection sensors are mostly incapable of giving a reliable measurement of objects accelerations [28], the target object acceleration $a_{t}$ is neglected at any time. The $d_{r}$ and $v_{r}$ are calculated from a pair of projection points over the tracked route, this functionality is supposed available in system failure condition.

### 4.3. Vehicle Motion Control

The velocity references provided by the rear-end collision-avoidance system are considered when objects are detected ahead instead of the trajectory references from the real-time planner. The vehicle motion control follows the references based on an MPC strategy. To perform this task, the model implemented by the MPC is a kinematic bicycle model (Section 3.1) with additional equations for $j_{x}$ and lateral error distance ($e_{y}$) to constraint it and assure an accurate lane-keeping,
(8)$$\left[\begin{array}{c} \dot{X}\\ \dot{Y}\\ \dot{\psi }\\ \dot{\delta }\\ \dot{v_{x}}\\ \dot{a_{x}}\\ \dot{e_{y}} \end{array}\right]=\left[\begin{array}{c} v_{x}\cos\left(\psi +\beta \right)\\ v_{y}\sin\left(\psi +\beta \right)\\ v_{x}\cos\left(\beta \right)\tan\left(\delta \right)/L\\ \Delta \delta \\ a_{x}\\ j_{x}\\ v_{x}\sin\left(\psi +\beta -\psi^{ref}\right) \end{array}\right]$$
where the $\psi^{ref}$ is the yaw angle reference for a current position of the vehicle over the route.

The state parameters η = [*X*
*Y*
ψ
$v_{x}$] and the control inputs *u* = [Δδ
$j_{x}$] are optimized for the whole *H*. The velocity reference is defined by the rear-end collision system when an object ahead is present, in other cases this reference comes from the planned trajectory as well as those for lateral vehicle motion control. The control inputs are after integrated to reproduce the steering and pedal position as actuation signals for the vehicle interface.

## 5. Case Study

In this section, the case study to evaluate the fail-operational approach is presented. First, the test scenario is defined based on a route in a real urban scenario. Secondly, the test platform to perform the DDT fall-back strategy is detailed. Finally, the parameters considered from the database and for the decision and control are mentioned.

### 5.1. Realistic Scenario

The realistic scenario considered to validate the proposed approach is a highly automated driving bus to carry passengers at the port of Malaga city (Spain), as depicted in Figure 6.

The selected test route covers a challenging environment with static objects in addition to difficult vehicle motion maneuvers as roundabouts, merging streets and intersections, as seen in Figure 6a.

In case of system failures, the ADS must respond without driver intervention to achieve a minimal risk condition bringing the vehicle to a safe state. In this sense, permitted and non-permitted stops are considered to be portrayed in Figure 6b avoiding to instantly stop.

The test case analyzed in this work is delimited to the evaluation zone depicted in Figure 6c. The failure to be studied is the possible malfunction of the GNSS position receiver (which is a vital part of the ADS), which starts degrading up to total failure and a dynamic driving task (DDT) fall-back strategy must be activated by the ADS.

As an additional issue, the case is considered in which the emergency shoulder cannot be used, as another vehicle is already parked, requiring driving of a longer distance to the next permitted stop while performing dead reckoning. Three different failure scenarios are analyzed, as shown in Figure 7.

When a failure of the GNSS occurs, the distance required ($d_{r}^{req}$) to achieve a safe-parking is calculated constantly before to initiate the maneuver as presented in the Equation (Equation 9). If the $d_{r}^{req}$ is lower than $d_{r}$ from and object and the available emergency shoulder longitude, then the lane-change maneuver initiates to achieve a minimal risk condition, parking the vehicle on the emergency shoulder. Moreover, the first one of the two terms in the right-hand side of Equation (Equation 9) can be employed to estimate a stop on-lane if permitted. On the contrary, the vehicle continues to the next permitted stop.
(9)$$d_{r}^{req}=v_{x}\left(\frac{v_{x}}{a_{x}^{degr}}+t^{delay}+t^{timeout}\right)+\frac{d_{ey}-e_{y}}{v_{ey}}$$
where $t^{delay}$ and $t^{timeout}$ are additional times considered to complete the lane-change maneuver being conservative. The $t^{delay}$ is stated as 0.5 s and related to actuation devices and vehicle’s inertia that retard the final stopping time. The $t^{timeout}$ is defined as 1 s considering a required time for the vehicle to be located enough on the emergency shoulder before totally stop.

It should be noted that the case study has been implemented in a simulation environment. This allows introduction of the degrading behavior in the perception system and evaluating the proposed fall-back strategies with minimal risk before future implementation.

### 5.2. Test Platform

A standard electric bus has been selected as the test platform for the case study scenario. This bus weights 16,000 kg and has a dimension of approximately 12.16×3.30×2.55 m, with a wheelbase of 5.77 m, a minimum turning radius of 7.2 m and a maximum front-wheel angle of 0.68 rad.

This way, the test platform has been modeled in Dynacar simulator [36], which uses a multi-body formulation to link the chassis with a steering knuckle suspension at the front axle, and a rigid axle suspension type at the rear. The suspensions are also linked to the two wheels at the front axle and four wheels at the rear axle, based on a standard Pacejka tire model defined in [37].

Moreover, the sensors have been simulated from the data obtained from the Dynacar model, introducing measurement errors to simulate degraded scenarios. The exteroceptive sensors can cover 360${}^{\circ }$ around the vehicle, reaching a maximum radius of 60 m for object detection. In the case of the GNSS sensor, which is the focus of the work, a random Gaussian noise associated with the quality signal of the GNSS-INS interface is added around the nominal state parameter obtained from the simulated test platform. The random noise values are introduced considering the quality of the signal to be simulated, as in real commercial devices (Table 2). Future implementation will need some of these exteroceptive sensors, the instrumentation necessary for real vehicles is detailed in [29].

### 5.3. Parameters

To test the proposed approaches, the following parameter values have been applied.

The noise covariances for the UKF have been calculated as suggested by [11], the process noise covariance matrix ($Q_{n}$) is defined assuming the standard deviation of parameters from the test vehicle circulating in normal conditions helps to determine $Q_{n}$. The measurement noise covariance matrix ($R_{n}$) is selected by taking into account the accuracy of commercially available acquisition devices. The $Q_{n}$ and $R_{n}$ are depicted in the Table 2.

The fail-operational positioning system requires the estimation of the Cornering Stiffness coefficients. As detailed in Section 3.1.2, a set of open-loop tests is carried out to determine the steady-state circular driving behavior described in [32], obtaining a set of data that can be used to create a cornering stiffness map.

For that purpose, the open-loop tests must cover the whole range of $v_{x}$ and δ for the test platform detailed in Section 5. Hence, the steering angle has been modified from −0.5 to 0.5 rad, in 0.1 rad steps, while the longitudinal speeds have taken the values of 0.5, 1, 2, 3, 4, 5 m/s.

The resulting cornering stiffness map is shown in Figure 8. From this map, intermediate values required by the fail-operational positioning system are estimated using interpolation.

In the case of the Real-Time trajectory planner, the start distance $d_{start}$, longitudinal velocity and deceleration in degraded mode are fixed to 5 m, 1.5 m/s and 0.2 m/s${}^{2}$, respectively, while ($v_{ey}$ and $d_{ey}$) are fixed for the case study proposed to 0.2 m/s and 4 m, respectively.

To implement the vehicle motion MPC controller, the following parameters have been used. A prediction horizon of H=10 is defined with a constant time step of 0.5 s. The states parameters and control input weights for optimization are intuitively defined as $Q_{w}$ = diag([1 1 25 1]) and $R_{w}$ = diag([10 10]), respectively, giving more importance to the vehicle orientation over the route. The physical constraints for state parameters and control inputs are summarized in Table 3.

Finally, it should be noted that the open-source ACADO Toolkit is employed to solve the optimal control problem both in the rear-end collision-avoidance system and vehicle motion control. A continuous output Implicit Runge–Kutta integrator of second-order simulates the system 1 integration step in both cases. The *H* is parametrized to obtain 10 elements with a constant time step of 0.5 s.

## 6. Results and Discussions

In this section, the most relevant results associated with the proposed fail-operational positioning system and the defined fall-back strategies are analyzed. Moreover, the effect of the proposed approach in the comfort of the passengers is also analyzed.

### 6.1. Evaluation of the Fail-Operational Positioning System

The robustness of the control architecture is evaluated here performing complete laps on the test circuit. The Figure 6a,b shows the route defined for the evaluation of the fail-operational positioning system based on UKF. This trajectory is executed using the control architecture proposed in Section 2. Four different scenarios are proposed, with different GNSS positioning qualities (from 2 to 5).

In each simulation, the positioning data gave by the raw GNSS-INS sensor (with Gaussian noise), the output of the UKF filter and the real position of the vehicle are measured, and compared with the trajectory reference, to calculate the lateral positioning error $e_{y}$ (m).

Results for the four signal quality scenarios are shown in Figure 9, where the statistic distribution of the lateral positioning error $e_{y}$ (m) is calculated considering the raw GNSS-INS sensor data (raw), the UKF filter output (UKF) and the real position of the vehicle (real).

From the results, the decrease in the quality of the GNSS signal increases significantly the lateral error if the raw data is used (raw case). It could be fatal in an automated vehicle operation such as the one analyzed. Moreover, the errors could introduce instability in the controllers, depending on the nature of the noise. This emphasizes the need for providing robust solutions to positioning measurements in automated vehicles.

Results also demonstrate the positive performance of the proposed UKF approach (UKF case), which can reduce in more than 90% the error associated with $e_{y}$ in the poorest quality condition (GNSS 2). This demonstrates the validity of the proposed approach. In addition, the level of performance that can be achieved using the UKF is demonstrated in the real position of the vehicle (real case).

### 6.2. Evaluation of the Dynamic Driving Task Fall-back Strategy

In this section, the proposed fall-back strategy performance is evaluated in the three different scenarios depicted in Figure 7: stopping before a parked vehicle, after a parked vehicle and continuing to a next permitted stop due to no space availability.

In all three scenarios, the same GNSS failure sequence will be evaluated, as depicted in Figure 6c. At the beginning of the test, the GNSS system has a higher signal quality, sequentially reducing it until a total failure exists. At that point, the fall-back strategy will have to take on the control of the automated bus and lead it to a minimum risk position using the data provided by the virtual positioning controller.

Figure 10 indicates, for each scenario, the fall-back sequence carried out. In the first row, the point in which the failure occurs (the same in the three cases) is depicted. In the second one, the activation of the degraded condition is shown, in which the speed of the vehicle is reduced. Then, when a free parking spot is activated, the lane-changing maneuver is activated, to finally brake and stop. Please note that the black lines represent the road borders, the green dotted line is the central line of the road, while the red and violet lines are the executed and calculated trajectories.

The performance data in the three scenarios are shown in Figure 11. In this graph, the vertical dashed lines define the starting points of the failure, degraded, maneuver and brake phases (stop is considered the end of the graph). Moreover, four main performance indicators are analyzed for each scenario. In the first row ($v_{x}$) the longitudinal speed reference given by the trajectory planner (*Reference*) and the real speed of the vehicle (*Ego-vehicle*) is depicted. In the second one ($d_{x}$), the longitudinal distance to the nearest object (parked vehicle) (*ObjectDistance*) and to the next emergency shoulder (*SpaceAvailable*) is shown. These distances are calculated with the position of these items in the planned trajectory. Also, the longitudinal distance required for performing the lane-change maneuver is shown (*SpaceRequired*). This calculation is detailed in the Equation (Equation 9). In the third row, the time evolution of the lateral error $e_{y}$ for the planned trajectory is shown, considering the raw data provided by the GNSS system (which fails) (*raw*), the output of the UKF (*UKF*) and the real position of the vehicle (*real*). Finally, the computational cost of the high and low-level controllers is shown.

From these graphs, several conclusions can be drawn. First, the robustness of the proposed UKF-based position estimator is demonstrated in all scenarios. If $e_{y}$ is analyzed, it can be noted that the effect of GNSS quality degradation directly affects the noise of the positioning system, which causes important $e_{y}$ errors (up to 1 m). However, as previously analyzed, the use of the proposed UKF-based estimator reduces the effect substantially.

Second, the proposed fail-operational Positioning System proves an effective approach in a total failure case. When total failure happens (black vertical dashed line), the data provided by the GNSS remains constant and no longer can be used to estimate the position. At this point, the Positioning Monitor of the fail-operational positioning system switches to the Virtual Positioning Controller, entering degraded mode while making use of the odometer and the steering wheel to estimate the position of the vehicle. Please note that due to the nature of the selected sensors, estimation errors in $e_{y}$ graphs will accumulate in time (see *real*), creating a drift. This effect is better seen in the third scenario, in which the nearest emergency shoulder is not available (is full) and therefore, the vehicle needs to move to the next one, operating more time in degraded mode. Hence, the degraded mode is intended to be used in emergencies for limited amounts of time or small distances, which is a valid assumption in urban environments such as the ones analyzed in the case study.

Third, the longitudinal speed ($v_{x}$) and distance ($d_{x}$) shows that the proposed fall-back strategy performs properly using the data provided by the fail-operational positioning system. When the failure occurs (black vertical dashed line), the vehicle reduced its speed to 1.5 m/s in all cases, entering a degraded state (blue vertical dashed line) once constant speed is achieved.

In this state, the trajectory planner searches for available spaces on the next emergency shoulder. For that purpose, the planner calculates the required space for the emergency parking maneuver (*SpaceRequired*) which depends on the current velocity and compares it with the distance to the nearest object/vehicle parked (*ObjectDistance*) and the available emergency shoulder distance (*SpaceAvailable*). Only if both are higher than the required distance to maneuver, the trajectory planner modifies the original route to start the lane-change maneuver. Please note that the object detection distance limit is 50 m and that the emergency shoulder-distance limit detection is 60 m, hence higher distances are limited to the maximum value.

The first scenario (parking before an object/vehicle in a shoulder), is the simplest one. It can be seen that when the degraded state is activated (97 s), the required space is less than the available shoulder distance, and the distance to the next vehicle, activating the lane-change maneuver (which implies a peak in $e_{y}$ due to the lateral reference change), and moving through the shoulder until the maneuver has been completed. In the second scenario (parking after an object/vehicle in a shoulder), the degraded state is activated at the same time, but in this case, the emergency shoulder is still available, but a vehicle is already parked and the relative distance to it is too low to maneuver. Hence, the vehicle continues moving until the parked vehicle is surpassed (115 s). At this point, 50 m of emergency shoulder remains, which is more than the space required for the maneuver. In the third scenario (no space), there are two vehicles parked in the emergency shoulder. Hence, when the degraded state is activated, the distance to the first vehicle, and then, to the second, is detected (the vehicle change is shown as a peak at time 115 s). When the second vehicle is surpassed, however, the remaining shoulder distance is not enough to maneuver safely, and the vehicle continues moving to the next emergency shoulder.

Finally, if the computational cost graphs are considered, it can be seen that the proposed approach is computationally efficient, requiring less than 1ms to execute in an Intel Core i7-6600u CPU, 2.60GHz and 16GB RAM. This demonstrates that the approach can be implemented in real time.

#### Evaluation of Passenger Comfort

Comfort is a key issue when considering automated driving solutions. Traditionally, comfort has been related to the magnitude of the lateral and longitudinal accelerations, being higher ones less comfortable for passengers.

In Figure 12 the lateral and longitudinal accelerations associated with the three scenarios analyzed in the previous section are depicted. Two situations are considered, the first row depicts the acceleration results when a degraded GNSS quality (level 2) exists when the failure happens. The second situation considers the case in which an optimal quality (level 5).

As can be seen, even before the failure, the differences in the lateral acceleration are important due to the noise that the GNSS presents in lower qualities. Lateral accelerations are an order of magnitude higher in these cases, resulting in more uncomfortable driving. Therefore, sensor quality directly can affect passenger comfort.

Please note that the longitudinal acceleration is not affected in this case, due to the odometry is used to estimate it. When failure occurs, similar behavior is achieved. However, the vehicle speed and accelerations are reduced when the positioning monitor switches to the virtual positioning sensor.

## 7. Conclusions and Future Works

Although the research and development in automated driving has considerably helped the implementation of higher SAE automation levels, current control architectures rely on the driver as a backup in case of system failures. Moreover, hardware redundancy is the usual action plan to ensure fail-operational systems, as few software solutions exist in the literature.

The present work targets the issue of the vehicle bringing itself to a safe state in degraded mode after a major failure in the position receiver. Instead of a progressive deceleration on the current lane, the system focuses on seeking a permitted space on the route, performing a lane-change maneuver to the emergency shoulder, and then executes a safe stop.

The fail-operational control architecture and systems proposed here are explained in depth. They include basic ADS features to achieve a minimal risk condition along a route, according to a realistic case study presented for bus shuttling services as: fail-operational positioning system, real-time trajectory planner, collision-avoidance system and vehicle motion controller.

The fail-operational positioning system comprises a UKF to improve the vehicle location due to lack of quality in the position receiver, an issue very common in urban scenarios where the satellite line-of-sight would be constantly obstructed. A virtual sensor switches on by a positioning monitor in case of total failure in the position sensor is detected, then a DDT fall-back strategy is possible performing dead reckoning with database information. A previous cornering stiffness estimation through open-loop tests provides useful information for the vehicle model employed.

The real-time trajectory planner is capable of comfortably slow-down the velocity reference after the failure, expecting an available and permitted space to perform a lane-change maneuver and safely locating the vehicle on the emergency shoulder. The benefits of having an object parked in advance are considered, hence the available space to initiate the parking maneuver is contrasted constantly with a required space calculation. A rear-end collision-avoidance system is activated at all times adapting the velocity reference to remain a safe distance to objects ahead.

Both the collision-avoidance system and the vehicle motion controller are based on MPC. It is possible to optimize the trajectory bearing in mind safety and comfort in maneuvers. The vehicle motion controller includes a lateral position restriction aiming to improve the lane-keeping performance, being possible to enhance it on one side when lane-change maneuvers are required avoiding that the vehicle goes out the road boundaries.

As this paper was focused on presenting a fail-operational control architecture approach in case of positioning failures, future works will consider in depth the maximum time–distance travel capacity in dead-reckoning circumstances under the influence of real instrumentation and the urban scenario presented in this article.

## Figures and Tables

**Figure 1 sensors-20-00442-f001:**
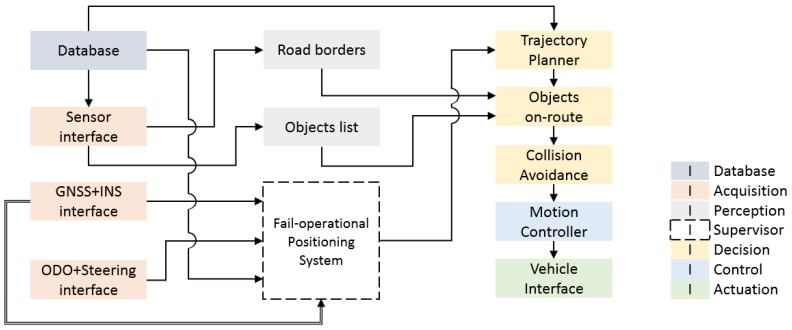
Control Architecture.

**Figure 2 sensors-20-00442-f002:**
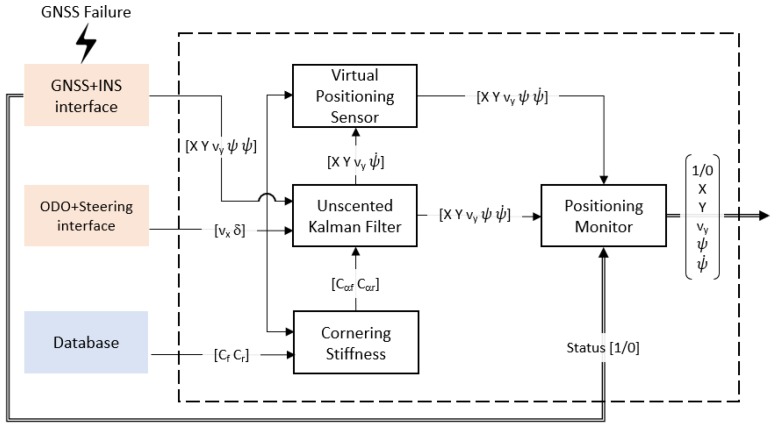
Flowchart on fail-operational Positioning System.

**Figure 3 sensors-20-00442-f003:**
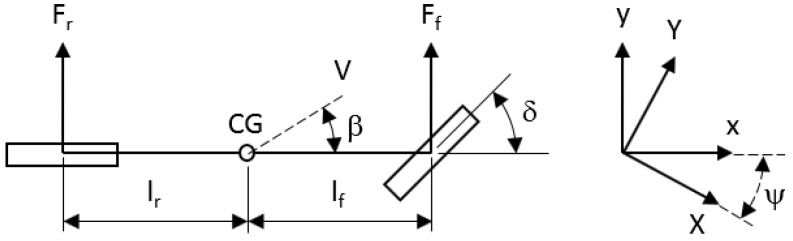
Simplified bicycle model for lateral dynamics.

**Figure 4 sensors-20-00442-f004:**
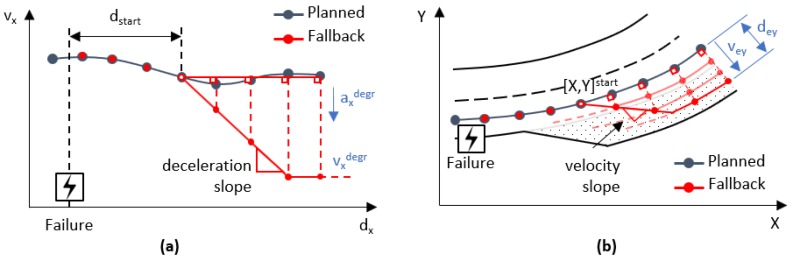
Real-time trajectory planning for (**a**) velocity and (**b**) path.

**Figure 5 sensors-20-00442-f005:**
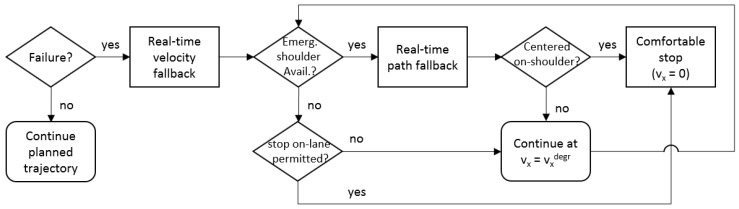
Flowchart on real-time trajectory planning.

**Figure 6 sensors-20-00442-f006:**
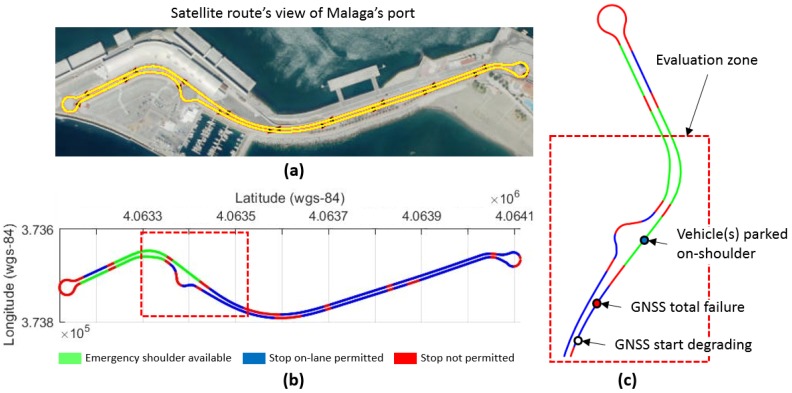
Realistic environment scenario for automated driving system tests on simulation. (**a**) Satellite’s view of urban route, (**b**) permitted and non-permitted stops in case of total positioning failure, and (**c**) evaluation zone for test case study.

**Figure 7 sensors-20-00442-f007:**
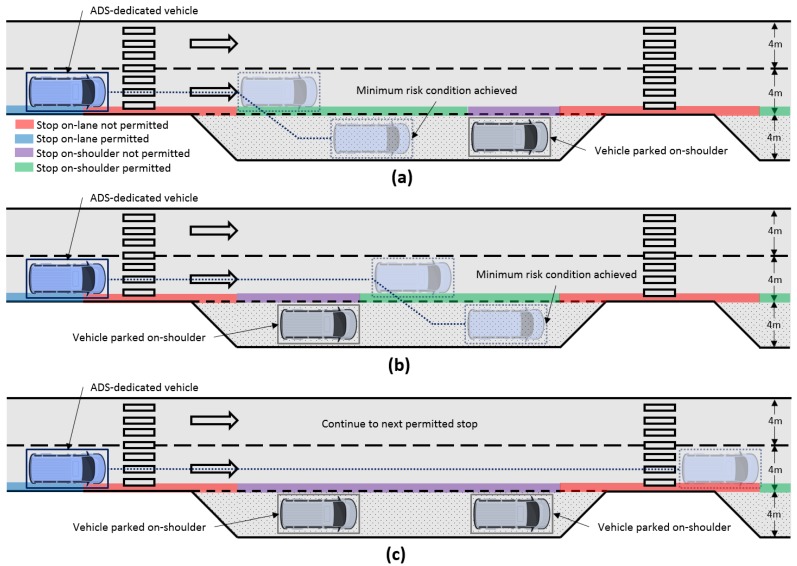
DDT fall-back strategy response under three different scenarios. A minimum risk condition is achieved (**a**) before and (**b**) after an object parked on the emergency shoulder. A next permitted stop necessary due (**c**) no space available on current emergency shoulder.

**Figure 8 sensors-20-00442-f008:**
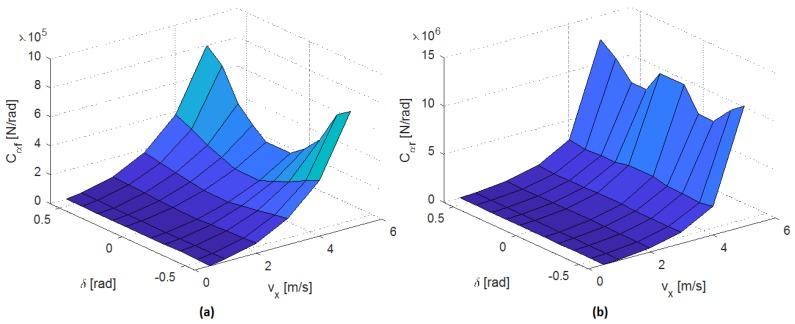
(**a**) Front and (**b**) rear cornering stiffness estimation

**Figure 9 sensors-20-00442-f009:**
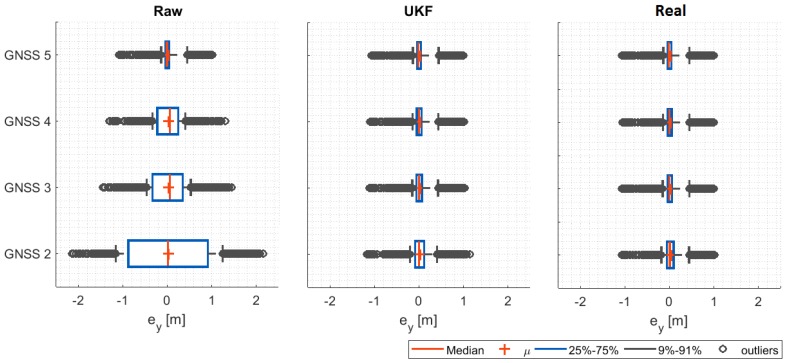
Lateral error under different GNSS positioning quality.

**Figure 10 sensors-20-00442-f010:**
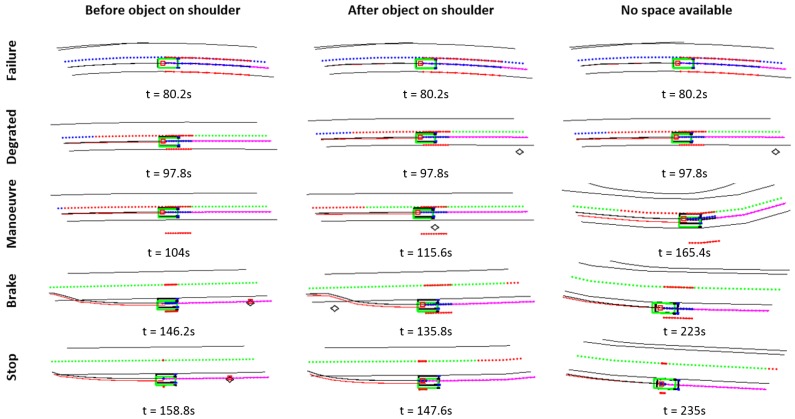
DDT fall-back strategy under different use cases.

**Figure 11 sensors-20-00442-f011:**
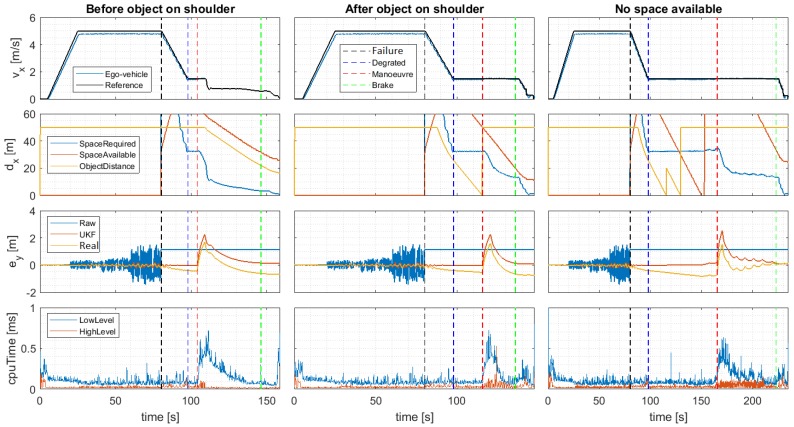
DDT fall-back response due GNSS total failure after continuously degrading position.

**Figure 12 sensors-20-00442-f012:**
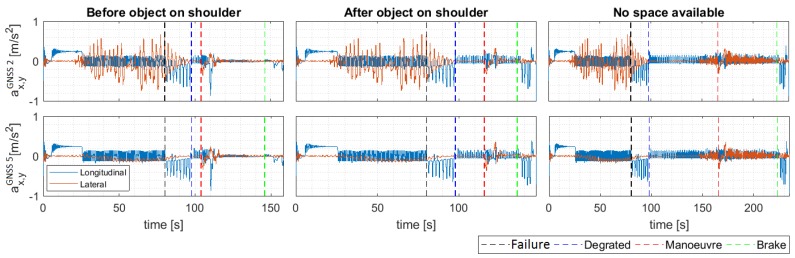
Longitudinal and lateral accelerations in DDT fall-back strategy.

**Table 1 sensors-20-00442-t001:** DDT fall-back strategies due automated driving system failures.

System	Functionality	Fall-Back Strategy
Perception	Object detection	Create ghost vehicles to replace the hidden ones due high curvatures in highways [15].
		Create ghost objects due sensor failure and perform lane-changing maneuver to emergency shoulder [16].
Decision	Lane centering	switch to differential braking control if electrical power steering fails [17].
	Trajectory planning	Emergency maneuver bring vehicle to stop if collision free trajectories fails [18].
		Emergency trajectory to stop at the slowest lane [19].
Control	Speed profile	Use a future velocity if communication of control messages or the propulsion controller fails [20].
	Collision avoidance	Brake if reception of data packets or inter-vehicle distance from lead vehicle fails [21].
Actuation	Drive-by-wire	Various forms of monitoring and redundancy are considered in failure cases [22].

**Table 2 sensors-20-00442-t002:** Process and measurement covariances in UKF.

Position Covariances					Inertial Covariances			
**Parameter**	**Quality**	$Q_{n}$	$R_{n}$	**Unit**	**Parameter**	$Q_{n}$	$R_{n}$	**Unit**
$\sigma_{XY}^{2}$	5	$1\times 10^{-3}$	0.0141	m	$\sigma_{y}^{2}$	$1.26\times 10^{-2}$	$2.78\times 10^{-2}$	m
	4		0.2828	m	$\sigma_{v_{y}}^{2}$	$1.26\times 10^{-4}$	$2.78\times 10^{-4}$	$\frac{m}{s}$
	3		0.4243	m	$\sigma_{\psi }^{2}$	$2.7\times 10^{-1}$	$1.7\times 10^{-1}$	$\frac{rad}{s}$
	2		1.1314	m	$\sigma_{\dot{\psi }}^{2}$	$2.7\times 10^{-3}$	$1.7\times 10^{-3}$	$\frac{rad}{s^{2}}$

**Table 3 sensors-20-00442-t003:** Constraints in the low-level control.

Parameter	Lower	Upper	Unit
Δδ	1	1	$\frac{rad}{s}$
$j_{x}$	1	1	$\frac{m}{s^{3}}$
δ	−0.68	0.68	rad
$v_{x}$	0	$v_{x}^{ref}$	$\frac{m}{s}$
$a_{x}$	$-a_{x}^{ref}$	0.2	$\frac{m}{s^{2}}$
$e_{y}$	$e_{yL}^{ref}$	$e_{yR}^{ref}$	m

Where $v_{x}^{ref}$ are the velocity references, $a_{x}^{ref}$ depends to the longitudinal operation mode defined in Section 4.2, and the $e_{yL}^{ref}$ and $e_{yR}^{ref}$ are the left and right lateral error distances to the borders, respectively, according the current position of the vehicle on-lane.

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
