# Peer review of "A Fail-Operational Control Architecture Approach and Dead-Reckoning Strategy in Case of Positioning Failures"

_sensors, 2020, doi:10.3390/s20020442_

Round 1
Reviewer 1 Report
In this paper a fail-operation control architecture is used for a very specific drive situation.
1) For lateral velocity, a simplified bicycle model is used. Why not use a complete model?
2) Pacejka tire model is used for the simulation. How are the Pacejka parameters of the tires obtained? How does the error in these parameters affect the control scheme?
3) Why are the cornering stiffness coefficients used in this simulation? How are these coefficients obtained?
4) Apparently, this work has not been simulated in real test conditions. No detail is given about the instrumentation needed in the real vehicle. If only simulated data is used, how one can be sure that the reality uncertainties are covered?
5) The test scenario is quite limited to a specific situation. To understand the reach and possibilities of the proposed fall back control scheme it would be good to know how the errors or uncertainties in the used model affects the result of the simulation. As only simulation data is presented, how do model parameter error estimation affects the results in the simulation? Have the authors run a sensitivity study to prove the robustness of the control?
6) Results in this study cannot be double-checked as information or specific details about the implementation is limited.
Author Response
First of all, we appreciate the comments and suggestions received from the reviewers and the editor, which have improved the quality of the paper. We reproduce below the reviewers' comments and explain how they have been addressed in the revised version for all reviewers.

Reviewer 2 Report
It has significance to develop a fail-operational control architecture approach in case positioning failures, and the proposed method is approciate, but authors must explain the following question clearly:
(1) Why did not consider the case without GPS signal?
(2) Please illustrate Figure 9 more detailed?
(3) Why is there longer manoeuvre time for the case After object on shoulder than that of Before object on shoulder in Figure 10?
(4) The Vx value is rather low for test, why?
Author Response

(The authors gave the same response as above.)

Round 2
Reviewer 1 Report
My main concern about the paper is that there is no way to reproduce the results obtained by the authors, and therefore they cannot be double checked. Author answer to my points 5 and 6 are lousy. I leave the final decision to the editor.